# Statistical signature of subtle behavioral changes in large-scale assays

**Alexandre Blanc**[1,2*], **François Laurent**[1,2,3*], **Alex Barbier-Chebbah**[1,2], **Hugues Van Assel**[4], **Benjamin T. Cocanougher**[5,6,7], **Benjamin M.W. Jones**[5,6,7], **Peter Hague**[5,6,7], **Marta Zlatic**[5,6,7], **Rayan Chikhi**[8], **Christian L. Vestergaard**[1,2], **Tihana Jovanic**[9☯*], **Jean-Baptiste Masson**[1,2☯], **Chloé Barré**[1,2☯]

**1** Institut Pasteur, Université Paris Cité, CNRS UMR 3571, Decision and Bayesian Computation, Paris, France, **2** Epiméthée, INRIA, Paris, France, **3** Institut Pasteur, Université Paris Cité, Bioinformatics and Biostatistics Hub, Paris, France, **4** École Normale Supérieure de Lyon, UMPA, Lyon, France, **5** University of Cambridge, Department of Zoology, Cambridge, United Kingdom, **6** MRC Laboratory of Molecular Biology, Neurobiology Division, Cambridge, United Kingdom, **7** Janelia Research Campus, Howard Hughes Medical Institute, Ashburn, Virginia, United States of America, **8** G5 Sequence Bioinformatics, Department of Computational Biology, Institut Pasteur, Paris, France, **9** Institut des Neurosciences Paris-Saclay, Université Paris-Saclay, Centre National de la Recherche Scientifique, UMR 9197, Saclay, France

☯ These authors contributed equally to this work.
* aleblanc@pasteur.fr (AB); cbarre@pasteur.fr (CB); jbmasson@pasteur.fr (J-BM); tihana.jovanic@cnrs.fr (TJ)

**Data availability statement:** There are no primary data introduced in the paper. The last

## Abstract

The central nervous system can generate various behaviors, including motor responses, which we can observe through video recordings. Recent advances in gene manipulation, automated behavioral acquisition at scale, and machine learning enable us to causally link behaviors to their underlying neural mechanisms. Moreover, in some animals, such as the *Drosophila melanogaster* larva, this mapping is possible at the unprecedented scale of single neurons, allowing us to identify the neural microcircuits generating particular behaviors. These high-throughput screening efforts, linking the activation or suppression of specific neurons to behavioral patterns in millions of animals, provide a rich dataset to explore the diversity of nervous system responses to the same stimuli. However, important challenges remain in identifying subtle behaviors, including immediate and delayed responses to neural activation or suppression, and understanding these behaviors on a large scale. We here introduce several statistically robust methods for analyzing behavioral data in response to these challenges: 1) A generative physical model that regularizes the inference of larval shapes across the entire dataset. 2) An unsupervised kernel-based method for statistical testing in learned behavioral spaces aimed at detecting subtle deviations in behavior. 3) A generative model for larval behavioral sequences, providing a benchmark for identifying higher-order behavioral changes. 4) A comprehensive analysis technique using suffix trees to categorize genetic lines into clusters based on common action sequences. We showcase these methodologies through a behavioral screen focused on responses to an air puff, analyzing data from 280 716 larvae across 569 genetic lines.

version of the software used in this work is accessible at https://github.com/DecBayComp/Detecting_subtle_behavioural_changes, and we have archived our code at the time of submission on Zenodo (https://doi.org/10.5281/zenodo.14674576). An archive of the trained autoencoder model, along with the training data, is also registered on Zenodo (https://doi.org/10.5281/zenodo.14674575).

**Funding:** J-BM, CLV, CB and AB received funding from the INCEPTION project (PIA/ANR-16-CONV-0005) and the "Investissements d'avenir" program (PIA) managed by Agence Nationale de la Recherche ANR-19-P3IA-0001 (PRAIRIE 3IA Institute); CLV and AB from the ANR funding ANR-SiNCoBe (ANR-20-CE45-0021); and TJ from the ANR PIA funding ANR-20-IDEES-0002, ANR-17-CE37-0019-01, and ANR-NEUROMOD (ANR-22-CE37-0027). TJ and JBM also received funding from the European Union's Horizon 2020 research and innovation program under the Marie Sklodowska-Curie grant agreement No 798050. The funders had no role in study design, data collection and analysis, decision to publish, or preparation of the manuscript.

**Competing interests:** The authors have declared that no competing interests exist.

## Author summary

There is a significant gap in understanding between the architecture of neural circuits and the mechanisms of action selection and behavior generation. *Drosophila* larvae have emerged as an ideal platform for simultaneously probing behavior and the underlying neuronal computation. Modern genetic tools allow efficient activation or silencing of individual and small groups of neurons. Combining these techniques with standardized stimuli over thousands of individuals makes it possible to causally relate neurons to behavior. However, extracting these relationships from massive and noisy recordings requires the development of new statistically robust approaches. We introduce a suite of statistical methods that utilize individual behavioral data and the overarching structure of the behavioral screen to deduce subtle behavioral changes from raw data. Given our study's extensive number of larvae, addressing and preempting potential challenges in body shape recognition is critical for enhancing behavior detection. To this end, we have adopted a physics-informed inference model. Our first group of techniques enables robust statistical analysis within a learned continuous behavior latent space, facilitating the detection of subtle behavioral shifts relative to reference genetic lines. A second array of methods examines subtle variations in action sequences by comparing them to a bespoke generative model. Our suites combine both bayesian and frequentist methods. Together, these strategies have enabled us to construct representations of behavioral patterns specific to a lineage and identify a roster of "hit" neurons with the potential to influence behavior subtly.

## Introduction

Animals integrate external sensory inputs and their internal states to generate suitable motor responses. This involves different areas of the nervous system, ranging from areas underlying sensory processing and higher-order processing to those governing decision-making and motor control. Furthermore, animals frequently respond to stimuli with a sequence of actions requiring precise control of transitions between individual actions. Different animals may react differently to the same stimulus, and the same animal can respond variably to repeated stimuli. This probabilistic nature of responses implies complexity and stochasticity in the behavioral choice mechanisms. The neurobiological interactions among neurons that regulate the trade-off between action stability and variability and control transitions between actions remain only partially understood.

Identifying the neural substrates responsible for behavior generation and selection within the nervous system is crucial for advancing our understanding of these mechanisms. Until recently, this task was challenging due to the difficulty in simultaneously manipulating neuron groups and capturing the corresponding behaviors. Furthermore, linking the often subtle changes in behavioral sequences across multiple time scales to neuronal manipulations has been hampered by the need for very large datasets and the inadequacy of available statistical methods.

The past decade has witnessed significant advancements in connecting behaviors with neural computations. Notably, data-driven neuron-behavior mappings have been established for *Drosophila melanogaster* in both adult [1] and larval stages [2]. *D. melanogaster* presents an ideal model for such studies due to its sufficiently complex yet accessible nervous system, comprising roughly 10 000 neurons in larvae and 130 000 neurons in adults. The complete

synaptic connectomes for larval (full CNS connectome) and adult (brain) *D. melanogaster* have been fully mapped [3–5], providing detailed diagrams of neuronal connections. Additionally, the *D. melanogaster* genome has been extensively characterized, and the development of thousands of GAL4 lines facilitates precise genetic manipulation [6,7], down to the level of individual neurons.

The semi-transparent cuticle of the larva enables the application of optogenetic techniques to selectively and reproducibly activate or inactivate neurons across the entire nervous system during behavior [2,8]. Techniques such as the targeted genetic expression of tetanus neurotoxin (TNT) can also disrupt synaptic transmission in individual neurons or small neuron groups. High-throughput tracking with real-time segmentation capabilities allows for recording hundreds of thousands of larvae, with individual neurons or neuron groups being selectively activated or silenced, constitutively or reversibly [9].

Advances in machine learning [10,11] have recently complemented automated behavioral analyses, and supervised [12–19] and unsupervised methods [20–28] have been introduced alongside image feature-based approaches to identify behaviors. Some methods can be applied broadly to various experiments after an annotation phase, such as DeepLab-Cut [12] while others are more specialized and apply to one animal in a specified behavioral paradigm [29,30]. Supervised techniques aim to define behaviors based on external expertise, while unsupervised ones seek to have them naturally emerge, later undergoing post-hoc validation by experts. Overall, the success of these methods depends on the definition of behaviors, the amount of accessible data and their standardization, and the variability expected under the experimental protocol. Usually, these frameworks link sensory stimuli or targeted neural activation to their behavioral output and are associated with statistical testing to detect significant events.

The main challenges in analyzing larval behavioral recordings are linked to the significant deformability of their bodies, the low resolution of images imposed to allow large-scale screening, the multi-temporal scales of their behavioral dynamics, and the vast diversity of larval morphological characteristics across populations of several hundred thousand animals. In spite of these complications, both unsupervised [2,23] and supervised [31–33] approaches have been successfully applied, albeit with known limitations. In supervised approaches, in particular, ambiguities in larva behaviour prevent full consensus on behavioural ground truth. New experiments suggest that additional action categories may be required to properly describe larval behavior, such as its C-shape behavior before rolling [34,35]. Furthermore, the diversity of larvae lengths, speeds, variations in the recording time of the larva, and the inherent deformability of the larva body induce challenges in estimating classification errors. These ambiguities bias the identification of neurons of interest, i.e., neurons able to modify the larva behavior, towards the ones inducing large behavioral deviations.

In this paper, we develop new statistical tests allowing the detection of neurons inducing subtle changes in behavior. To ensure the robustness of such finer analyses, we first introduce a physics-informed Bayesian approach to regularize the recorded shapes of the larvae. We then introduce two statistical approaches to provide a global analysis of the larva behavioral screen and identify neurons able to induce subtle variations in local behavior or in the higher-order statistics of sequences of actions. We apply these approaches on an entire behavioral screen and demonstrate the ability of both approaches to detect neurons or groups of neurons that can induce subtle behavioral changes. We leverage our new approaches to provide compact representations of behavioral phenotypes and recast the behavioral phenotypic characterization problem into statistical testing procedures.

## Materials and methods

### *Drosophila melanogaster* stocks

The screen consisted of 569 GAL4 lines, as listed in S1 Table. These lines were from the Rubin collection lines (available from Bloomington stock centre), each of which is associated with an image of the neuronal expression pattern shown at flweb.janelia.org/cgi-bin/flew.cgi. In addition, we used the insertion site stocks, w;attP2 [6], OK107GAL4, 19-12-GAL4, NompC [36], and iav-GAL4 [37]. We used the progeny larvae from the insertion site stock, w;;attp2, crossed to the appropriate effector (UAS-TNT-e (II)) for silencing. The w;;attP2 were selected because they have the same genetic background as the GAL4 tested in the screen. We used the following effector stocks: UAS-TNT-e [38] and pJFRC12-10XUAS-IVSmyr::GFP (Bloomington stock number: 32197).

### Behavioral apparatus, experiments and screen design

**Apparatus** The setup was fully described previously [9,33] (Fig 1). Briefly, it consists of a video camera for monitoring larvae, a ring light illuminator, and custom hardware modules for generating air puffs, controlled through the multi-worm tracker (MWT) software [8,39].

**Behavioral experiments** The experiments are fully described in [9]. Briefly, they started with collecting embryos for 8–16 hours at 25 °C with 65% humidity. Larvae were raised at 25 °C with normal cornmeal food. Foraging 3rd instar larvae were used (larvae reared 72–84 hours or for three days at 25 °C). Before experiments, larvae were separated from food with 10% sucrose, scooped with a paintbrush into a sieve and washed with water. The substrate for behavioral experiments was a 3% Bacto agar gel in 25 625 cm$^2$ square plastic dishes. Batches of approximately 50 to 100 larvae were imaged in each behavioral assay. The larvae were left to crawl freely on an agar plate for 44 seconds before the stimulus delivery. The air puff was delivered at the 45th second and applied for 38 seconds. Two different stimulus intensities were considered, one at a high intensity of 6 m/s and the other at a lower intensity of 3 m/s. In this paper, when a result is stated without indicating a specific intensity, it should be understood that it was obtained with the higher 6 m/s.

**Screen design** The screen consisted of recordings of the behavior of 569 GAL4 lines from the Rubin GAL4 collection, where we constitutively silenced small subsets of neurons and individual neurons using tetanus toxin [33,38]. We selected these lines from the entire collection for sparse expression in the brain and ventral nerve cord of the larval CNS and expression in the sensory neurons. Of the 569 lines tested here, several neuronal lines were not part of the Rubin collection: we added 19–12-GAL4 and NompC-GAL4 for sensory neurons and OK107GAL4 for the mushroom body. Each GAL4 line was screened using the air-puff assay described above. This article used no activation method (optogenetic or other) since we used constitutive silencing.

**Behavioral dictionary** Six stereotypical actions are commonly used to constitute the behavioral dictionary of the larva (Fig 1B): A: crawl, B: bend (all turning actions), C: stop (not moving), D: hunch (fast retraction of the head) E: back crawl (crawling backwards), and F: roll (defensive manoeuvre consisting in sliding laterally). We use the letters A–F in plots and tables for brevity. Where these actions were needed for the analysis, we inferred them using the method introduced in [33].

### Physics-informed regularization of larva shape

We conducted large-scale imaging by recording larvae with a wide-field view, allowing us to analyze up to 100 larvae per plate. This approach is time-efficient but results in images of

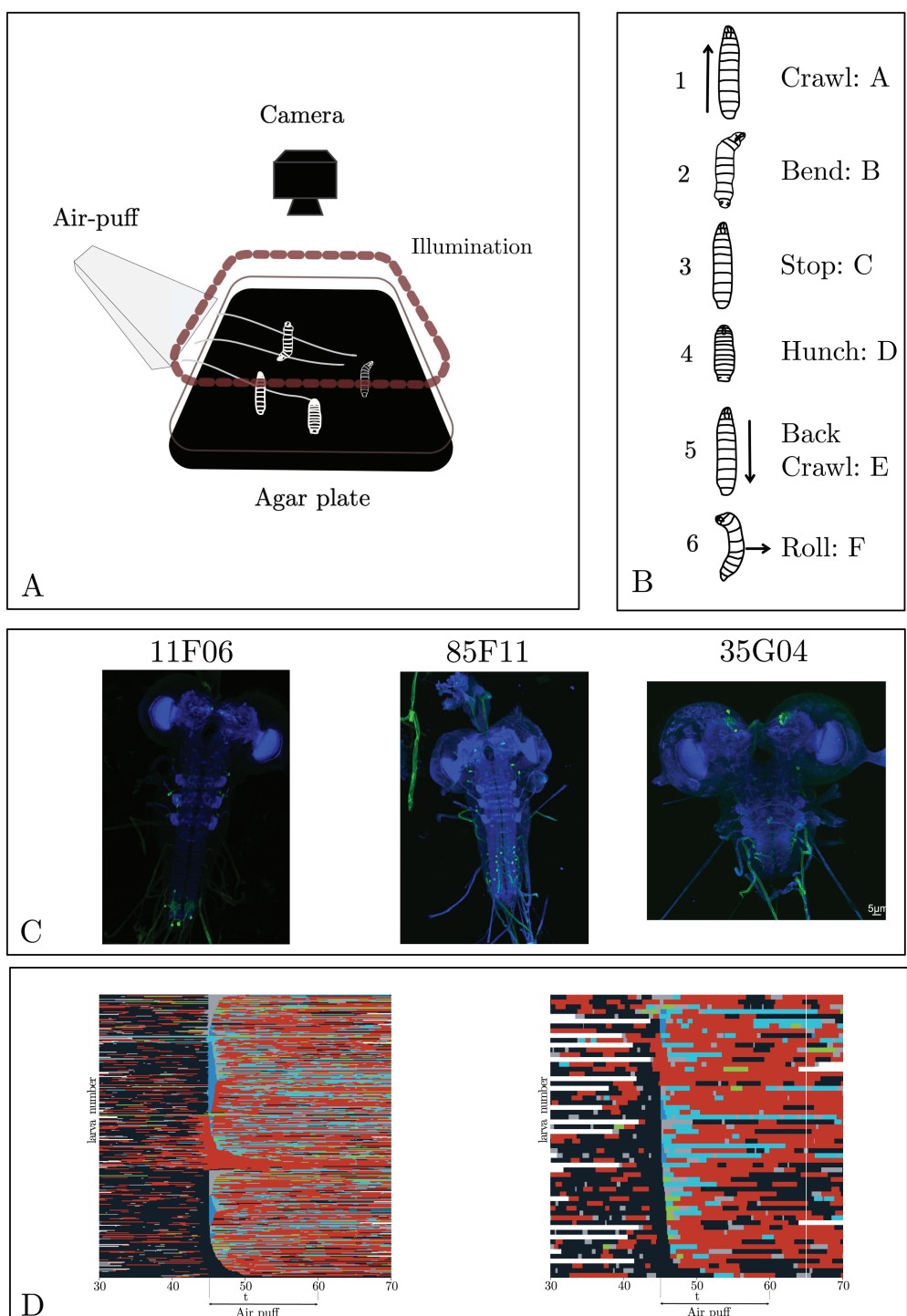

**Fig 1. (A) Behavioral set-up.** The larvae move freely on an agar plate, and their movement is recorded with an infrared camera equipped with a high-throughput closed-loop tracker. The stimulus was an air puff (or illumination for training data). (B) The six stereotypical actions [8,9] associated with the larva for this experimental paradigm. (C) Example of Neuronal expression patterns in three example lines: 11F06, 85F22, and 35G04. (D) Ethogram of larva behavior in response to an air-puff at 45s based on automated behavior detection. Each line corresponds to one larva, with the control line (attP2) on the left and R35G04 on the right. Colors correspond to the following actions: black for crawl, red for bend, blue for stop, deep blue for hunch, and cyan for back. Note that no rolls were observed in these lines.

lower resolution. Additionally, the large scale of our experiments meant that many larvae were not perfectly dried, leading to abnormal contour shapes. Impurities in the agar further contributed to these irregularities, as illustrated in Fig 2. Such contour abnormalities risk leading to misclassification of larva behavior, potentially introducing bias into subsequent statistical analyses. To ensure an accurate representation of larval shapes, we developed the following regularization procedure based on physics-informed Bayesian inference [40].

**Preprocessing** MWT extracts contours with a variable number of points in each frame, depending on the larva's size. We represent each contour by the points $f(i) = (x(i), y(i))$ for $i \in \{1, 2, \ldots, N_{\text{track}}\}$. We regularized the shape by fixing the number of point in the contour to $N_{\text{track}} = 50$ coupled with a low-pass filtering. In particular, we generated the contours by

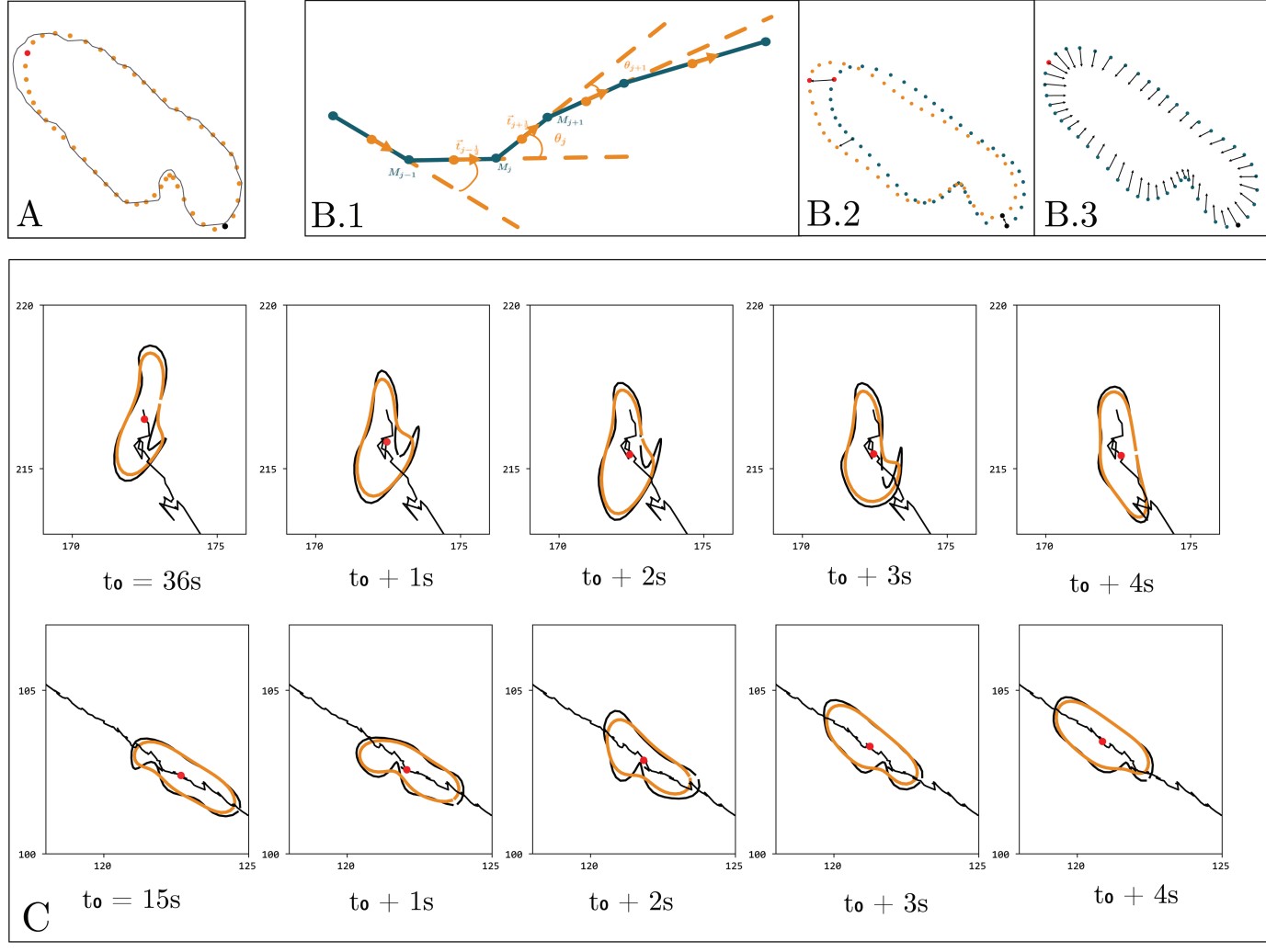

**Fig 2. (A) Noisy, tracked contour of a larva in gray and regularized contour in orange.** The head is indicated by a red point and the tail by a black. (B) 1. Close-up of six points of the larva contour. Vectors between these points represent the contour. The $j$th point is denoted $M_j$, its tangent vector $\vec{t}_j$, and the curvature at this point $\theta_j$. 2. Two larval outlines at time $t$ and $t + dt$; the vectors show the movement of two selected points during the time-lapse $dt$. 3. Change of the contour points after the surface energy is minimized. (C) Results of the algorithm applied to two different larvae at four different time steps with the tracked contour in black and the inferred one in orange. The larva's trajectory is drawn in black, and its center of mass is indicated by a red dot (see also S1 Video).

retaining the $K$ lowest Fourier modes [41] of the recorded contour (Fig 2A),

$$S_n = a_0(f) + \sum_{k=1}^{K} \left[ a_k(f) \cos\left( kn \frac{2\pi}{N_{\text{track}}} \right) + b_k(f) \sin\left( kn \frac{2\pi}{N_{\text{track}}} \right) \right], \quad (1)$$

with $a_k$ and $b_k$ the Fourier coefficients,

$$a_k = \sum_{i=1}^{N_{\text{tracking}}} \frac{f(i)}{N_{\text{track}}} \cos\left( \frac{2\pi ki}{N_{track}} \right), \quad (2)$$

$$b_k = \sum_{i=1}^{N_{\text{tracking}}} \frac{f(i)}{N_{\text{track}}} \sin\left( \frac{2\pi ki}{N_{\text{track}}} \right). \quad (3)$$

We reconstructed the shape $S$ by retaining the $K = 7$ lowest harmonics, a number chosen empirically to prevent discontinuities [41]. This initial reconstruction ensured screen-scale regularization of larvae contours regardless of their variability in size and shape.

**Simplified physics model of the larva** We designed a minimal, effective 2D physics model to approximate the dynamic shape of the larva. It models the larva as an elastic contour with an active membrane energy. The total energy of the larva is the sum of a kinetic energy, a surface energy, and a bending energy: $E = E_k + E_S + E_b$ (Fig 2B.2).

The first term, the kinetic energy, is given by

$$\begin{aligned} E_k &= \int_\Omega \frac{1}{2} \rho v(s)^2 ds, \\ &= \sum_{i=1}^{N} \frac{m}{2} \frac{[x_i(t) - x_i(t - dt)]^2 + [y_i(t) - y_i(t - dt)]^2}{dt^2}, \end{aligned} \quad (4)$$

where $\Omega$ is the surface of the contour, $\rho$ is the surface density, $v(s)$ is the speed of the contour in the point $s$, $m$ is the total mass of the membrane, and $dt$ is the time lapse between images. Here $m$ (or, equivalently $\rho$) is a free parameter (i.e., a hyperparameter) of the model.

The second term, the surface energy, is given by

$$\begin{aligned} E_S &= \int_\Omega K ds^2, \\ &= \sum_{i=1}^{N} K[(x_i - x_{i-1})^2 + (y_i - y_{i-1})^2], \end{aligned} \quad (5)$$

with $K$ the elastic modulus (a second hyperparameter of the model).

Finally, the bending energy is expressed as

$$\begin{aligned} E_b &= \int_\Omega 2k(C - c)^2 ds, \\ &= \sum_{i=1}^{N} 2k\theta_i^2, \end{aligned} \quad (6)$$

where $k$ is the bending modulus, $C$ is the mean curvature over the entire contour, and $c$ is the spontaneous curvature defined by $c\,\hat{\mathbf{n}} = \frac{d\hat{\mathbf{t}}}{ds}$ with $\hat{\mathbf{n}}$ the unit normal vector and $\hat{\mathbf{t}}$ the unit tangent vector of the contour in curvilinear coordinates. For a discrete point, this curvature equals $c_i = \theta_i$ (Fig 2B.1), and we set $C = 0$.

**Inference**  We used Bayesian inference to infer the larva's regularized shape $\Sigma = \{M_1', M_2', ..., M_N'\}$. Its posterior distribution is given by

$$P(\Sigma|S) \propto P(S|\Sigma)P(\Sigma), \tag{7}$$

where $P(S|\Sigma)$ is the likelihood of the model and $P(\Sigma)$ is the prior, which regularizes the inference by incorporating our physical model. The prior is given by

$$p(\Sigma) \propto e^{-[E_S(\Sigma)+E_k(\Sigma)+E_b(\Sigma)]}, \tag{8}$$

up to a normalizing constant that does not influence the inference. The likelihood enforces proximity between the recorded contour and the inferred one according to a quadratic loss function,

$$P(S|\Sigma) = e^{-\frac{\sum_i \|M_i - M_i'\|^2}{\lambda}}. \tag{9}$$

We set $\lambda = \frac{1}{2}$ with no loss of generality, as the absolute scale of the energy does not impact the inference.

The log-posterior distribution is thus given by

$$\log(p(\Sigma|S)) = E_S + E_k + E_b - 2\sum_i \|M_i - M_i'\|^2. \tag{10}$$

The model's hyperparameters ($m$, $k$, and $K$) were set to balance the contributions of the three energy terms, giving them similar weights. The larva's mass was set to $m = 1$, the curvature coefficient to $k = 1$, and the elastic modulus to $K = 5$. In the numerical implementation, we corrected spurious high curvature anomalies by capping the energy by replacing $E_k$ by $E_k^{\text{eff}} = \tanh(E_k/\sigma)$ with $\sigma = 100$. We used stochastic gradient descent [42] to infer the maximum a posteriori (MAP) regularized contour $\Sigma$. We show in Fig 2C inferred contours for two examples displaying significant anomalies. Note that while we rely solely on the MAP of the shape in downstream analysis and not on the full posterior distribution, it is accessible using Markov Chain Monte Carlo [42] sampling if necessary.

## A continuous self-supervised representation of behavior

We developed a continuous representation of the behavior based on self-supervised learning (SSL) to alleviate the need for a predefined behavioral dictionary to characterize larva actions. SSL is a general paradigm [43–49] in which a model is trained using auxiliary objectives to improve the performance for downstream tasks. The auxiliary objectives are generally constructed from the data themselves, thus requiring no external labeling. Here, we present our implementation based on larva positional prediction of the regularized shapes inferred as described above (Physics-informed regularization of larva shape).

**Architecture and training of the neural network**  We used an autoencoder architecture comprising an encoder and a decoder, mapping input and output data, as illustrated in Fig 3A. The encoder takes a sample, $X_t$, as input and maps it to a learned latent space, producing the sample's latent representation. The decoder takes as input a latent representation and generates a reconstruction of the sample, $\hat{X}_t$, in the data space. We augment the objective by requiring the decoder to also predict the preceding and following samples,

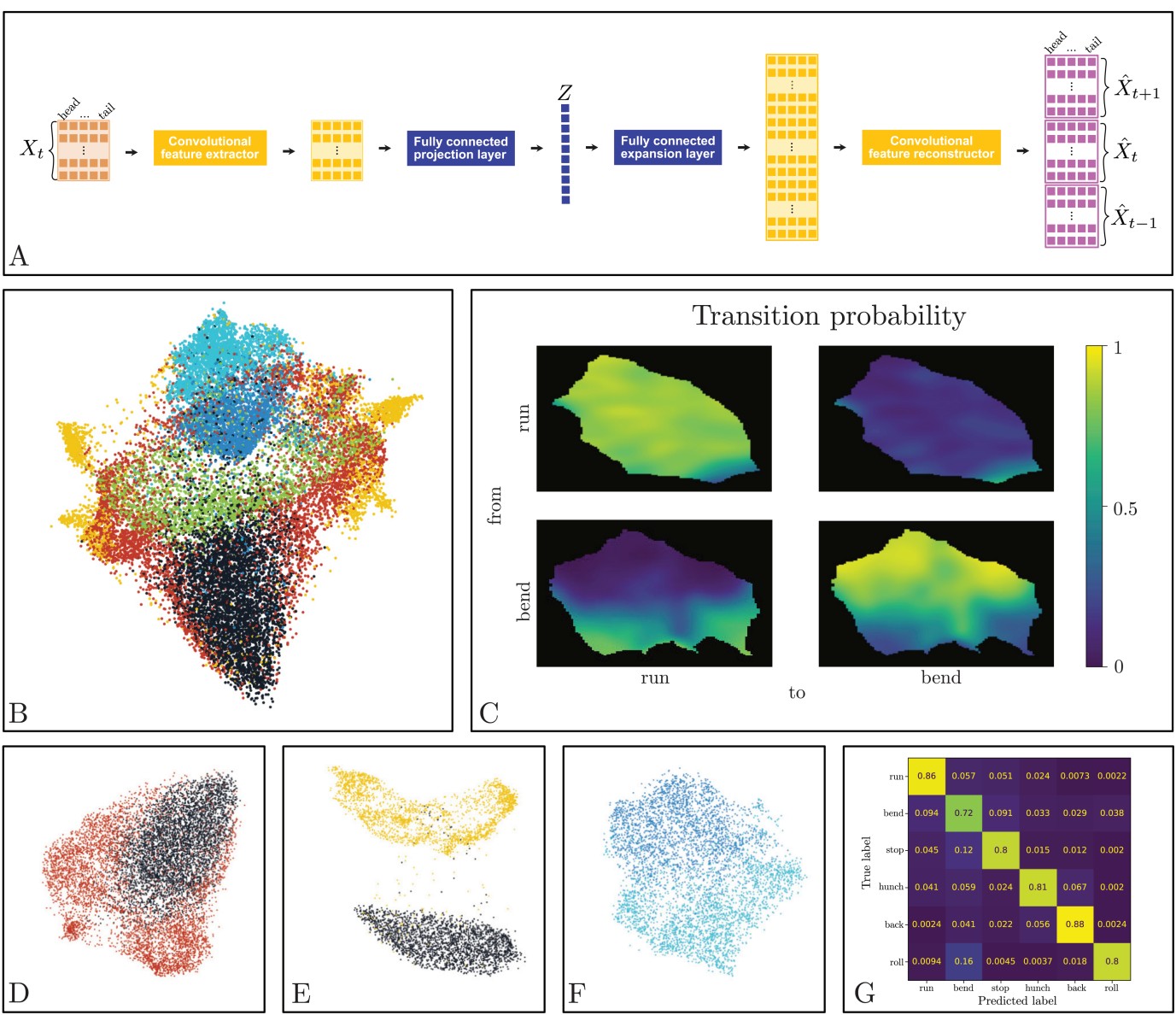

**Fig 3. (A) Architecture of the self-supervised predictive autoencoder.** The encoder consists of multiple convolutions with ReLU activations alternating between the spatial and temporal axes of the data, followed by a fully connected linear layer. The decoder consists of an upsampling linear layer matching the internal representation to the desired shape, followed by alternating convolutions with ReLU activations. (B) Visualization of the latent space. The 10D latent space is projected into 2D using UMAP [50]. The colors correspond to the discrete behavior dictionary (black: crawl, red: bend, green: stop, blue: hunch, cyan: back, and yellow: roll) (C) Transition probability from one discrete state to another as a function of the position in the latent space: here, between run and bend. (D–F) Highlights of the behavior geometry in the latent space (represented in 2D using UMAP). In D run vs. bend, in E run vs. roll, and in F hunch vs. back. (G) Cross-validated confusion matrix of random forest classifiers using the latent representation to infer the usual discrete behavior dictionary.

$X_{t-1}$ and $X_{t+1}$, forcing the neural network to encode the temporal continuity of the larvae's motion.

We used an epoch approach to encode the dynamics of the larva with a predefined time interval of length $\tau$. Unless indicated otherwise, we used $\tau = 2s$. In principle, $\tau$ can range from very short durations (a few hundred ms), capturing primitive muscular responses, to longer durations (5-10 s), capturing entire sequences of actions. The choice of $\tau = 2s$ was informed by

previous behavioral analysis of larvae [8,9,33,51] and was chosen to ensure the identification of transitions between different actions. The autoencoder was tasked (Fig 3A) to encode the coordinates of the larva within the present epoch and to predict the coordinates for the immediate future and past epochs (of the same duration $\tau$). The autoencoder was trained to minimize the mean squared error of reconstructed sequences. The autoencoder performs both 2D convolutions and 1D convolutions, acting alternately on the spatial and temporal coordinates (Fig 3A). Hyperparameters, architecture, and source code are provided at https://github.com/DecBayComp/Detecting_subtle_behavioural_changes.

**Datasets for training and testing** Tracking data were initially generated using MWT [39] as described above. We post-processed them using the pipeline introduced in [8], limiting the representation of a larva to 5 points along its anteroposterior axis [8] (tail, lower neck, neck, upper neck, head) to allow the representation to be used with very low-resolution imaging of larvae (as in [52]).

We assigned one of the six following discrete behavioral categories to each time point of the larva: *run*, *bend*, *stop*, *hunch*, *back* and *roll* using the pipeline of [33]. To promote generalization across lines and robustness regarding different morphologies, larva coordinates were normalized such that the line average of the larva length before sensory stimulus was equal to one. Furthermore, we centered and aligned the larvae so that the trajectories were centered around the origin. Data were sampled from experiments in [2] and in [9,33,53,54]. To promote generalization across lines and robustness regarding different morphologies, larva coordinates were normalized such that the line average of the larva length before sensory stimulus was equal to one. Furthermore, we centered and aligned the larvae so that the trajectories were centered around the origin. Data were sampled from experiments in [2] and in [9,33,53,54].

Natural behavior statistics are deeply imbalanced. Before the sensory input signal, the animals move freely, with roughly 70% run and 30% bend with occasional stops, although rhe sensory stimulus can generate behaviors that would not be evoked without stimulation. All data pooled together, regardless of experimental protocols, exhibited the following statistics with Run: 50.28%, Bend: 39.35%, Stop: 6.43%, Hunch: 0.81%, Back: 3.06%, and Roll: 0.07%. Lexical approaches, such as the one from [21], while very efficient in analyzing animal behavior in natural settings, have challenges with such a level of imbalance.

We thus used an inductive bias to train our predictive autoencoder: We selected training data to consist of consist of 100 000 samples, 10% of which were held out for validation, with 25% runs, 25% bends, and 12.5% of each of the other four stereotypical actions.

## Genotype-level analysis

**Genotype representation** To detect genotypes of interest (commonly called *hits*), we employ a non-parametric statistical test within the latent behavior space, learned as described in the previous section (A continuous self-supervised representation of behavior). While the testing relies on the learned behavioral space, we emphasize that other architectures and objectives (such as [22]) may be used if they provide a sufficiently robust description of the behavior.

Following the stimulus, we embedded the $\tau$-long windows of behavior using the predictive autoencoder, resulting in a sample of behavioral responses represented as points in the latent space. One larva's behavioral dynamics becomes a singular point within the latent space, whereas a genotype's experimental behavioral dynamics, evaluated, for example, on 1000 larvae, becomes a distribution of 1000 points inside the latent space. We estimated the

underlying distribution using a Gaussian kernel. Therefore, the phenotypic characterization of a genotype reduces to a probability distribution in the learned latent space, allowing us to reduce the comparison of two genotypes to a comparison of two probability distributions in a low-dimensional space (Fig 4A).

**Kernel-based statistical testing** We used the maximum mean discrepancy [55] (MMD) to measure the distance between distributions (Fig 4B), which is efficient in detecting subtle differences between distributions of point data [56]. MMD was developed to perform non-parametric statistical testing between two sets of independent observations in a metric space $\mathcal{Z}$ (here, the latent space $\mathcal{Z} = \mathbb{R}^{10}$). We denote by $X = \{x_1, \ldots, x_m\}$ the first set, drawn from the distribution $p$, and by $Y = \{y_1, \ldots, y_n\}$ the second set, drawn from $q$. The goal is to test if $p = q$, i.e., we seek to reject the null hypothesis that the two genotypes have similar behavioral responses.

The MMD between two probability measures $p$ and $q$ is defined as

$$\mathrm{MMD}[\mathcal{F}, p, g] = \sup_{f \in \mathcal{F}} \left( \mathrm{E}_x \left[ f(x) \right] - \mathrm{E}_y \left[ f(y) \right] \right), \tag{11}$$

where $\mathcal{F}$ is a class of functions from $\mathcal{Z}$ to $\mathbb{R}$, and $\mathrm{E}_x$ and $\mathrm{E}_y$ denote expectation with respect to $p$ and $q$, respectively.

When the function class is the unit ball in a reproducing kernel Hilbert space $\mathcal{H}$, the square of the MMD can be estimated directly from data samples [57]. We estimated the squared MMD between $X$ and $Y$ using the unbiased estimator given by

$$\mathrm{MMD}_u^2[\mathcal{F}, X, Y] = \frac{1}{m(m-1)} \sum_{\substack{i,j \\ i \neq j}} k(x_i, x_j) + \frac{1}{n(n-1)} \sum_{\substack{i,j \\ i \neq j}} k(y_i, y_j) - \frac{2}{nm} \sum_{i,j} k(x_i, y_j), \tag{12}$$

where $k$ denotes the kernel operator, specifically a Gaussian kernel given by $k(x,y) = \frac{1}{\sqrt{2\pi}\sigma_{\mathrm{ker}}} \exp\left(-\frac{\|x-y\|_2^2}{2\sigma_{\mathrm{ker}}^2}\right)$. The bandwidth $\sigma_{\mathrm{ker}}$ was calibrated using the median of the pairwise distances in the latent space of samples corresponding to the reference line following [55].

The MMD framework provides explainability of the statistical test by enabling the identification of the variables that exhibit the greatest difference between datasets [57,58]. It defines a particular function over the vector space that supports the distributions, called the witness function, which highlights regions where large deviations occur (Fig 4C). These regions can be analyzed further to identify the behavioral features that are associated with them.

## Probabilistic generative model of action sequences

In addition to the dictionary-free approach for behavioral analysis developed above (A continuous self-supervised representation of behavior and Kernel-based statistical testing), we developed a structured probabilistic approach to probe higher-order behavioral patterns that directly influence action sequences. This approach compares each genotype to a constrained generative model with a behavior dictionary instead of relying on direct comparison to a reference genotype.

The sequences of actions are modeled using a time-varying continuous-time Markov chain, built upon simple probabilistic basis functions (similar to [59]). The model is parametrized by the average duration of each action, the action's amplitude (either the maximum asymmetry factor, which serves as an experimentally robust proxy for the bending angle, or the velocity of the action), and the transition probabilities between successive

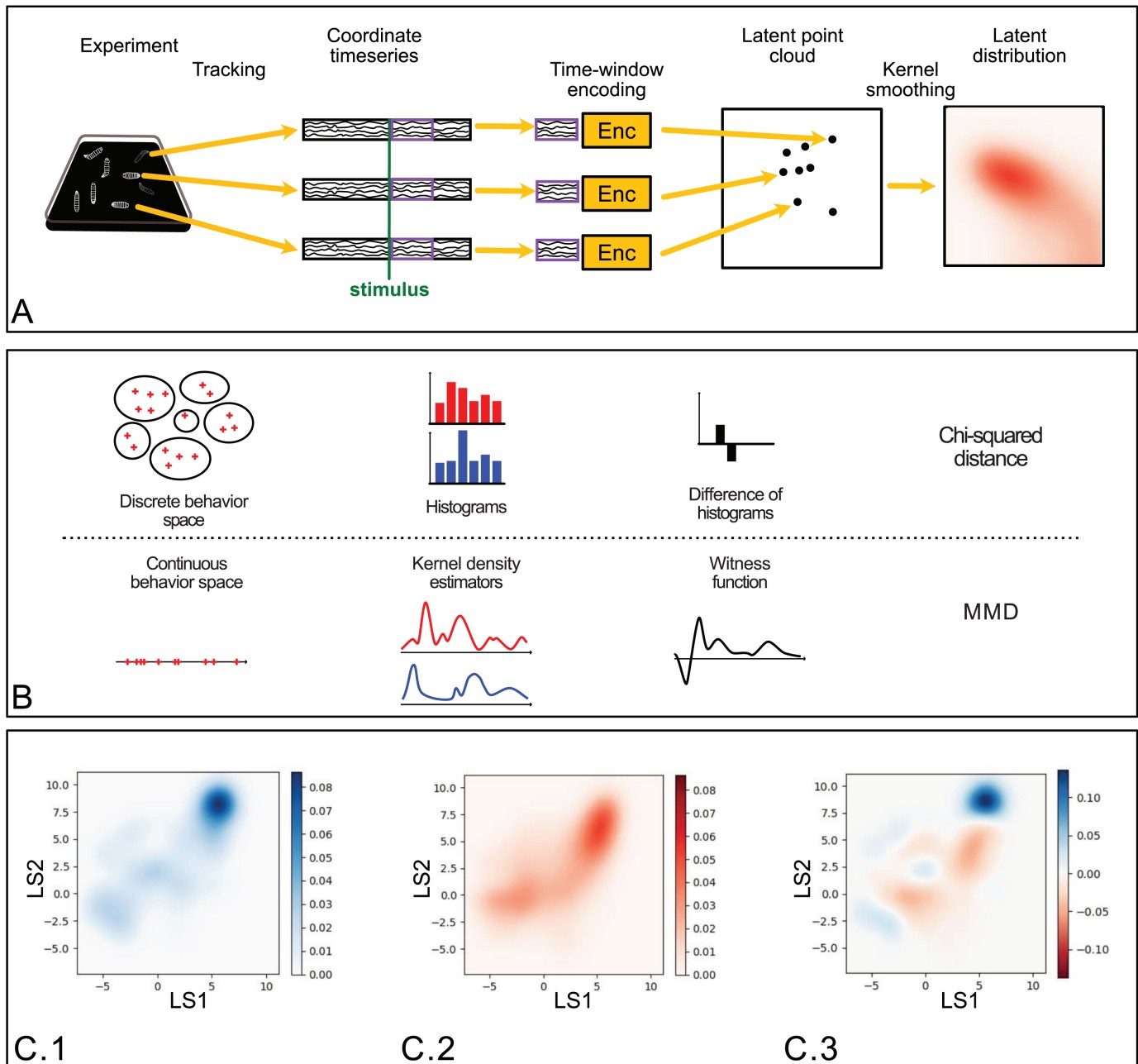

**Fig 4. (A) Illustration of our phenotyping modeling strategy for each genotype.** From left to right: The behavior evolution on the experimental setup is reduced to the five tracked points of the larva; the extraction of a temporal window (shown in purple on the ethogram as an illustration) usually after the onset of the stimuli (shown as a vertical green line), the projection of the temporal window on the latent space using the encoder shown in Fig 3 and reduced here to a yellow box, each point in the latent space corresponds to one larval behavior during the selected time window, and the phenotype of the genotype is the distribution of all the points in the latent space regularized by a Gaussian kernel. (B) Illustration of the correspondence between statistical testing procedures based on discrete behavior categories with chi-squared tests and testing procedures based on continuous behavior with MMD. (C) Latent distributions of behavior (regularized by a Gaussian kernel): (C.1) of the reference line *attP*2 and (C.2) of the line 10*A*11. (C.3) Witness function between these two latent distributions, highlighting the main behavioral differences between the lines.

actions. All parameters are allowed to vary temporally. We account for this time variation by using piecewise constant parameters in time windows of duration $\delta t = 1$ s.

We initialize the state of the larva following the stationary distribution of actions, $p(i, t_0)$. At a given point in time $t$ (including $t = t_0$), the duration $\Delta t$ of an action is drawn from a Poisson distribution,

$$p_i(\Delta t|t) = \lambda_i(t)e^{-\lambda_i(t)\Delta t}, \tag{13}$$

with $1/\lambda_i(t)$ representing the mean time spent in behavior $i$ at time $t$. The action's amplitude is allowed to depend on $\Delta t$. For asymmetric actions (i.e., bend, stop, and roll) the amplitude is quantified by the "asymmetry" factor $A_s$, while for the other actions (i.e., run, back, and hunch), it is quantified by the velocity $v$. The asymmetry is defined by $A_s = \frac{(1-\cos(\alpha_i))}{2}$, where $\alpha_i$ is the angle between the segment formed by the center and the head of the larva and the segment formed by the center and the tail of the larva. The asymmetry can take values in the range $[-1,1]$. We approximate the amplitude distributions using a kernel density estimator with a mixture of Gaussian kernels with uniform weights,

$$f_i(a_i) = \frac{1}{nh\sqrt{2\pi}} \sum_{j=1}^{n} e^{-\frac{1}{2}\left(\frac{a_i-\mu_{i,j}(t)}{h}\right)^2}, \tag{14}$$

where $a_i$ is the amplitude of the velocity or the asymmetry of the behavior $i$, $\mu_{i,j}(t)$ is the mean of $j$th component of the mixture, $h = \left(\frac{4\sigma^5}{3n}\right)^{\frac{1}{5}}$ is the bandwidth of the kernel, and $\sigma$ is the standard deviation of the amplitudes inferred from data. We set $n = 10$, which empirically leads to a good trade-off between bias and variance. Thus, the distribution is described by a 10-dimensional vector $\mu_i(t) = (\mu_{i,1}(t)(t), \dots, \mu_{i,10}(t))$. Finally, each new action is chosen according to a first-order Markov chain parametrized by the transition matrix $T(t)$. Since we explicitly model action durations, self-transitions are not possible, and the diagonal elements of $T(t)$ are thus equal to zero.

The full set of parameters to infer is $(\mathbf{\Lambda}, \mathbf{M}, \mathbf{T})$, where $\mathbf{\Lambda} = \{\lambda_i(t)\}$ is the expected inverse durations of each behavior in each time window; $\mathbf{M} = \{\mu_i(t)\}$ are the features for each action during each time window conditioned on the duration of the action, with $\mu_i(t, \Delta t) = (\mu_{i,1}(t)(t), \dots, \mu_{i,10}(t))$; and $\mathbf{T} = \{T(t)\}$ is the set of transition matrices in each time window.

The model's parameters are learned from experimental data using Bayesian inference (see Eq( 7)). The likelihood for one larva's behavior sequence can be written as

$$\mathcal{L}(\mathbf{X}|\mathbf{\Lambda}, \mathbf{M}, \mathbf{T})$$
$$= \prod_{s=0}^{t_{\text{end}}-\delta t} \prod_{m=1}^{M} \left[\lambda_{i_m}(s)e^{-\lambda_{i_m}(s)\Delta t_m}\mathbf{T}_{i_m,i_{m+1}}(s)f(a_m|\mu_{i_m}, dt_m)\right]^{\chi_{[s,s+\delta t)}\left(\sum\limits_{p=0}^{m-1} dt_p\right)}, \tag{15}$$

where $\mathbf{X} = \{(i_1, a_1, \Delta t_1), \dots, (i_M, a_M, \delta t_M)\}$ is the sequence of the larva's actions, $t_{\text{end}}$ is the duration of the recording, $\chi_{[a,b)}(x)$ is the indicator function for $x$ being in the interval $[a,b)$, and $i_m$ is the $m$th action, with $\Delta t_m$ its duration and $a_m$ its amplitude.

We regularize the inference using the following priors on the temporal variation of the parameters:

- A prior enforcing the normalization of the transition matrix:

$$\pi(\mathbf{T}) = e^{-\beta\left(\sum_{j\neq i} \mathbf{T}_{i\to j}-1\right)^2};$$

- A prior reinforcing a smooth temporal variation of

$$\Lambda : \pi(\Lambda) = \prod_i \prod_s e^{-\gamma(\|\lambda_i(s) - \lambda_i(s + \delta t)\|^2)};$$

- A prior reinforcing a smooth temporal variation of

$$\mathbf{M(t)} : \pi(\mathbf{M}) = \prod_i \prod_s e^{-\gamma(\|\mathbf{M}_i(s) - \mathbf{M}_i(s + \delta t)\|^2)},$$

The maximum a posteriori values of the parameters are inferred by minimizing the following cost function:

$$F = -\sum_{n=1}^{N} \left[ \log(\mathcal{L}(\mathbf{X_n}|\Lambda, \mathbf{M}, \mathbf{T})) + \log(\pi(\Lambda, \mathbf{M}, \mathbf{T})) \right], \tag{16}$$

where $\pi(\Lambda, \mathbf{M}, \mathbf{T}) = \pi(\Lambda)\pi(\mathbf{M})\pi(\mathbf{T})$ and $n \in \{1, 2, ..., N\}$ indexes the different larvae. We minimize this function with a direct gradient descent algorithm on the entire set of behavior sequences of the larvae of a single genotype. We visually represent the model in Fig 5A.

After inference, we generate artificial behavioral sequences from our model using Monte Carlo sampling of the posterior distribution. We then create new artificial sequences using the procedure outlined in Algorithm 1 in S1 Text.

We evaluated the model's goodness of fit using the MMD in the learned latent space (A continuous self-supervised representation of behavior and Kernel-based statistical testing) by comparing generated sequences with real sequences of behaviors. For each line, we took groups of 100 random larvae for which we calculated the probabilities of sequence occurrence. We obtained a distance for each line corresponding to the differences between the models and the experiments. We provide the distances S2 Table.

We additionally supplemented the global scoring of the MMD with sequence-based scoring, allowing a direct comparison of the probability of a defined sequence under the fitted model and its experimental frequency. To do this, we used the Z-score $Z = \frac{x - \nu}{\sigma}$, with $x$ the probability of the sequence under the model, $\nu$ the frequency of the sequence in experimental data, and $\sigma$ a bootstrap estimation of the standard deviation of the sample frequency under our generative model. We limited the analysis to sequences of 3 actions to maintain statistical significance for most lines in the screen. Fig 5B shows an example of Z-scores for all sequences for two different lines. Although the model reproduces the evolution of probabilities over time, some sequences on line 38$H$09 are poorly described, as evidenced by their large Z-score (the values of the Z-scores per sequence for selected lines are noted in S3 Table (during stimulus) and S4 Table (over the whole recording)).

## Clustering behavioral sequences from suffix tree representations

The total number of actions performed during this screen is roughly 1.3 million. By analyzing the screen as a general ensemble, the scale and diversity of recorded behaviors can be exploited to identify higher-order temporal structures in the behavioral sequences. We used a suffix tree representation to explore the entire screen and the genetic lines' organization. We constructed the suffix tree with Ukkonen's algorithm [60,61] on all sequences of behavior of all larvae across all genetic lines. We considered only the sequences of categorical behavioral actions, regardless of their durations. Each sequence was added to the suffix tree (see Fig 6A for an illustrative example of a tree built from just three different larvae). The size of the tree

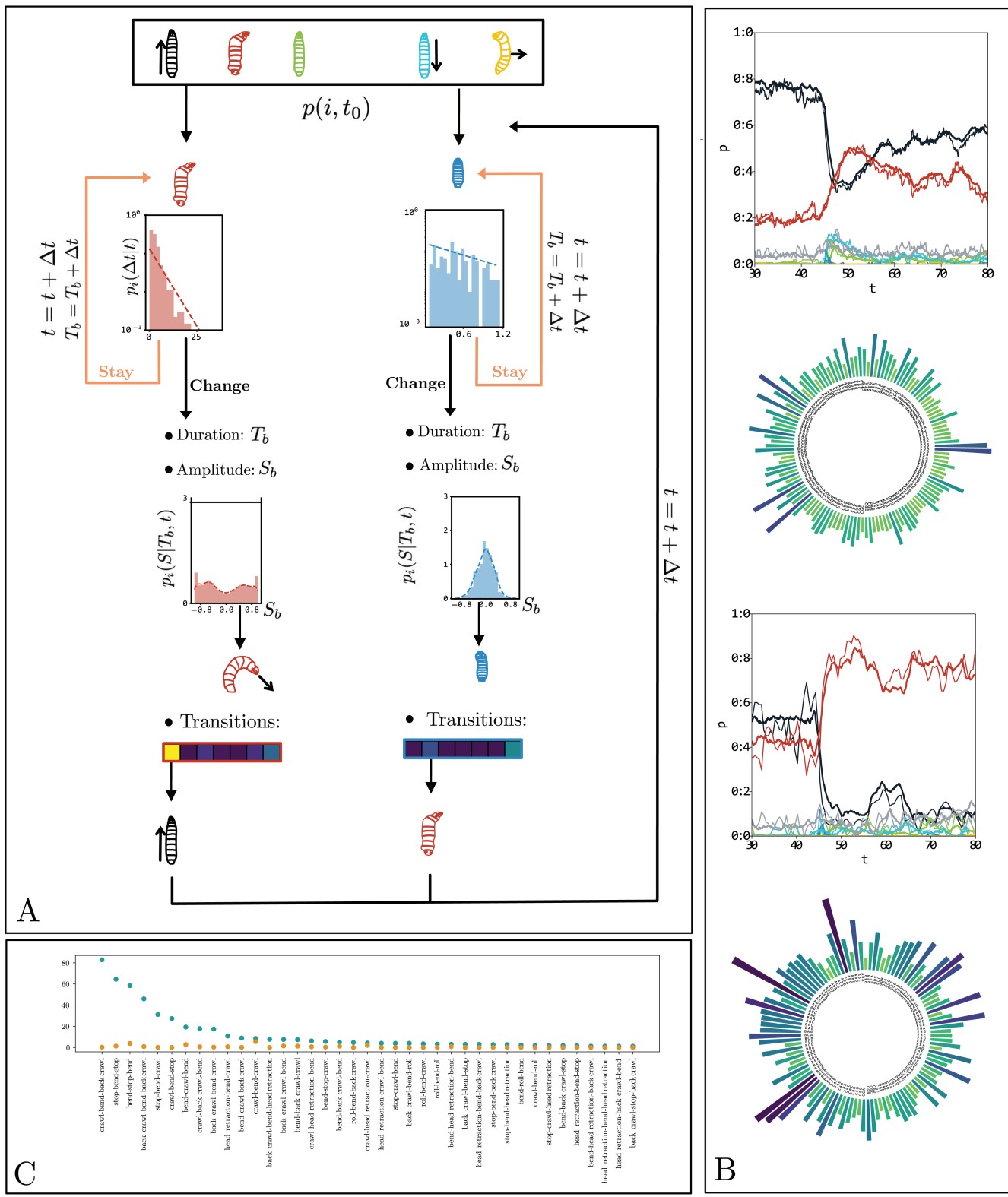

**Fig 5. (A) Graphical representation of the probabilistic generative model, showing the temporally inhomogeneous Poisson model $p_i(\Delta t|t)$, the distribution of action amplitudes $p_i(S|T_b, t)$, and transition probabilities to the other actions.** (B) characterization of behavioral responses to an air puff with the prediction of the generative model for two lines. At the top: time evolution of the larva's actions; thin lines represent the experimental recording, and thick lines are the generative model. At the bottom, a circular plot of the z-scores between the action sequences of the generative model and the experimental recordings. Darker blue colors indicate larger values. The two lines are $R41D01$ on top and $R38H09$ on the bottom.

grows quadratically with the length of the sequence, and the proportion of sequences common to several lines decreases accordingly. Thus, to avoid too long sequences (which would decrease statistical power due to their multiplicity), and to focus on the biologically relevant immediate response behaviors, we only considered behaviors occurring during the first 5 seconds after the onset of the stimulus. This has the additional benefit of limiting the computational burden.

We used the suffix tree to cluster different lines by utilizing internal nodes shared between several lines. The advantage of using a suffix tree is the ability to compare sequences of different lengths while retaining the order of behaviors. In particular, we compared each node of the suffix tree, examining node overlaps across multiple lines in the suffix tree. To define a metric, we embedded the behavioral sequences of each genetic line using the vector space document (VSD) model [62]. In this case, a document corresponds to a genetic line. We mapped the nodes from the common suffix tree to an $M$-dimensional space in the VSD model.

In the VSD model, each genetic line, $\ell$, is considered as a vector in an $M$-dimensional term space. We constructed these vectors by the term frequency-inverse document frequency (tf-idf) weighting scheme proposed in [64,65]. It measures the relevance of a word (here, a node) according to its frequency in each document (genetic line) and in a document collection (cluster of lines). The vector corresponding to a line $\ell$ is given by

$$\vec{\ell} = \{\omega(1, \ell), \omega(2, \ell), ..., \omega(M, \ell)\}, \tag{17}$$

where $\omega(i, \ell)$ is the weight applied to each node $i$ in document $\ell$, defined by the tf-idf,

$$\omega(i, \ell) = \log[1 + \text{tf}(i, \ell)] \log[(1 + N/df(i)]. \tag{18}$$

Here $N$ is the number of lines, $\text{tf}(i, \ell)$ is the frequency of the $i$th node in the line $\ell$, and $df(i)$ is the number of lines containing the $ith$ node. The frequency is given by $\text{tf}(i, \ell) = \frac{n_i}{N_\ell}$, where $n_i$ is the number of larvae that pass through this node $i$ and $N_\ell$ is the number of total larvae in the line $\ell$. We obtain a vector for each line, with a weight term for each node. We calculated a squared distance matrix from these vectors containing the pairwise distances between the vectors of the genetic lines. We measured the distance between two vectors $\vec{\omega}$ and $\vec{\omega}'$ as the cosine similarity since it is more robust to the variability in the number of larvae per line compared to the Euclidean distance. The cosine similarity is given by

$$d(\vec{\omega}, \vec{\omega}') = 1 - \frac{\vec{\omega} \cdot \vec{\omega}'}{\|\omega\| \|\omega'\|}. \tag{19}$$

We finally applied hierarchical clustering to the distance matrix to group the genetic lines according to their behavior sequences (Fig 6B).

## Results

We applied our methods to infer behavioral phenotypic descriptions that allow to characterize behavioral phenotypes of *Drosophila* larva both at the global scale (across the entire screen) and at the local scale (at the level of individual genotypes). To achieve a geometric perspective on the relationships between various genotypes at a large scale, we constructed a behavior distance matrix encompassing all lines by analyzing the distribution of genetic lines within the latent space. Our approach involved a two-step process: calculating the MMD distance

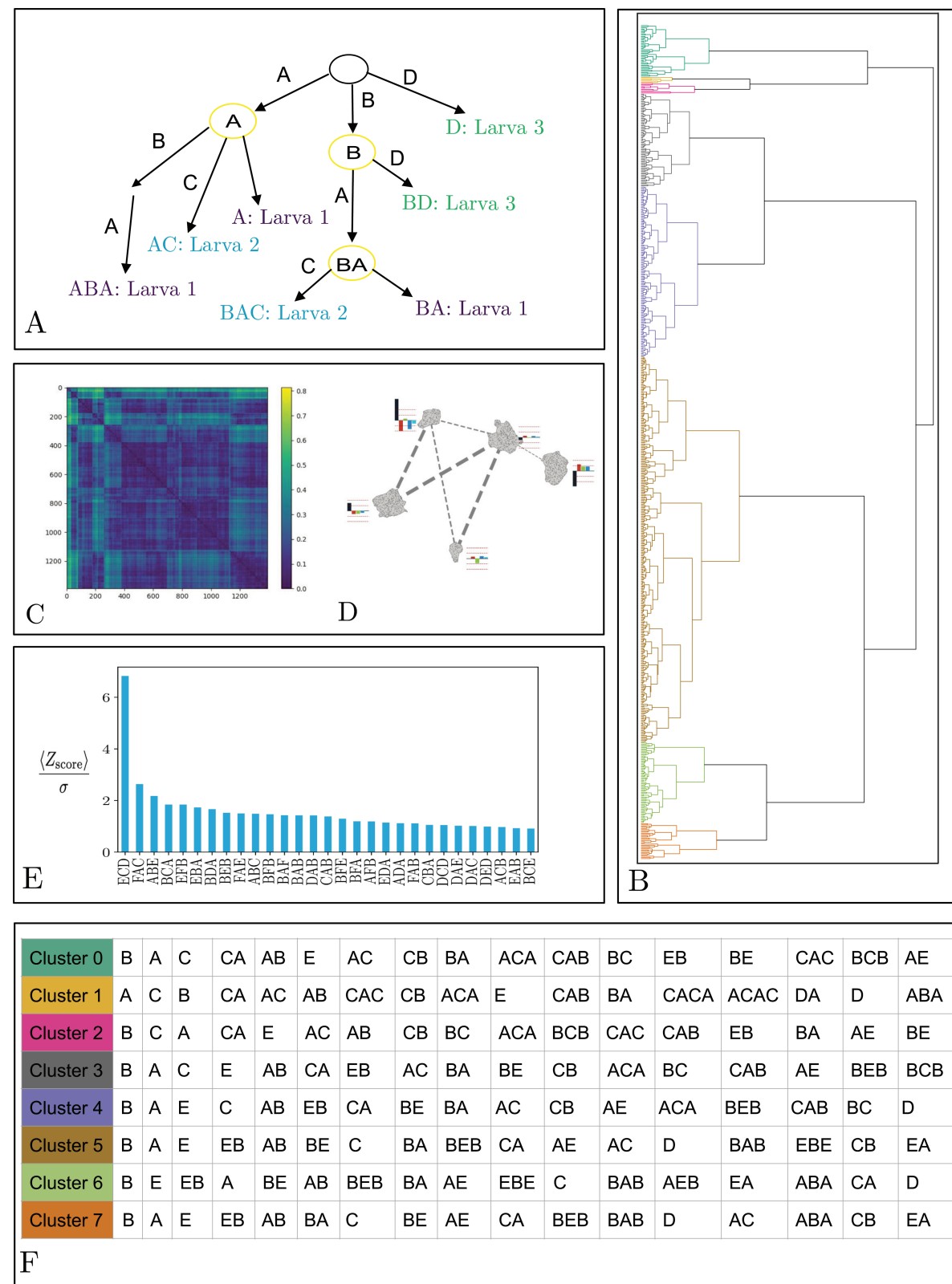

**Fig 6. A. Illustrative example of a suffix tree obtained from three larvae performing three different sequences.** Larva 1: ABA, Larva 2: BAC, Larva 3: BD, the seven paths from the root to the leaves correspond to the seven suffixes: A, BA, ABA, AC, BAC, D and BD. Each node shared by at least two larvae is shown in circles: A, B and BA. **B.** Hierarchical clustering based on the cosine similarity between the

suffix tree vectors of each genetic line. Each color is associated with a different cluster. C. Distance matrix representing the squared MMD between all lines from the inactivation screen, computed in a 10D learned latent space for a 2-second time window. D. 2D representation of the geometric relationships between lines, obtained using supervised UMAP [63], encoded by the distance matrix. The bar plot associated with each cluster represents the average variation of behavior during the 2-second window in the six actions behavior dictionary. The thickness of the lines linking two clusters is associated with the coupling between the clusters. E. The z-score distributions' standard deviation normalises average z-scores between data and generated sequences. We display only the 30 highest values. F. The 17 sequences of nodes with the highest frequency of occurrence for each of the eight clusters.

matrix for all lines and then embedding the genotypes into a high-dimensional geometric space through multidimensional scaling. This space, another latent space, represents the probabilistic reactions to stimuli at the genetic line level. We visualised this latent space using hierarchical clustering with Ward's linkage method (Fig 6D), identifying five contiguous regions. For a conventional representation of the primary behavior statistics, we calculated and compared the average behavior histograms for lines within each region against the overall average behavior histogram derived from this geometric framework. Additionally, we explored the interrelations among these regions using supervised uniform manifold approximation and projection (UMAP) for dimensionality reduction [63]. This process resulted in each subregion being represented as a separate connected component in a 2D space. We also illustrated the total connectivity between each region, quantified by the sum of edge weights in the graph created through the UMAP algorithm, shown in dashed gray lines with widths proportional to the logarithm of the connectivity (Fig 6D).

The scope of the screen, combined with the variety of genetic lines and behaviors, facilitates the categorization of larval dynamics into clusters and sequences of behaviors. By employing hierarchical clustering on the representation vectors of nodes within the suffix tree, which captures both the frequency of a sequence's occurrence and the number of lines displaying it, we can depict the principal families of larval behaviors in response to this sensory stimulation paradigm (Fig 6F). This approach reveals the larvae's reactions to airflow natural stimuli, notably bending movements. A distinction emerges between response families characterized by hunch (head retraction) and repetitive transitions between back movements and bends, and those characterized by rapid escape involving running phases, stopping phases, and then swiftly resuming running and bending cycles. The latter represents the baseline behavior of larvae placed on a 2D agar plate without a specific task, likely engaging in foraging behavior.

Our generative model (Probabilistic generative model of action sequences allowed us to identify behavioral sequences that are most likely unable to be described by a time-inhomogeneous Markov model (Fig 6E). These sequences, where back and hunch are often represented, are associated with avoidance manoeuvres. Numerous questions remain regarding how behavioral sequences are encoded and their neural generative implementation. Similarly, encoding the duration of these behavior motifs needs to be investigated, as some lines, for example GMR_18A10_AE_01@UAS_TNT_2_0003, may exhibit one of back or bend for a long time after the stimulus without transition to other behaviors, while other lines, for example GMR_12F10_AE_01@UAS_TNT_2_0003, may exhibit rapid transitions between the two.

A main result of our work is a catalog of genetic lines exhibiting subtle behavioral modifications, as identified through statistical testing within the behavioral latent space and via our Bayesian generative model for action sequences. The genetic lines pinpointed by our methodologies are detailed in the Supplementary Information S6 Table. Selected examples are shown in Fig 7, which presents the reference line alongside two instances of lines identified using the

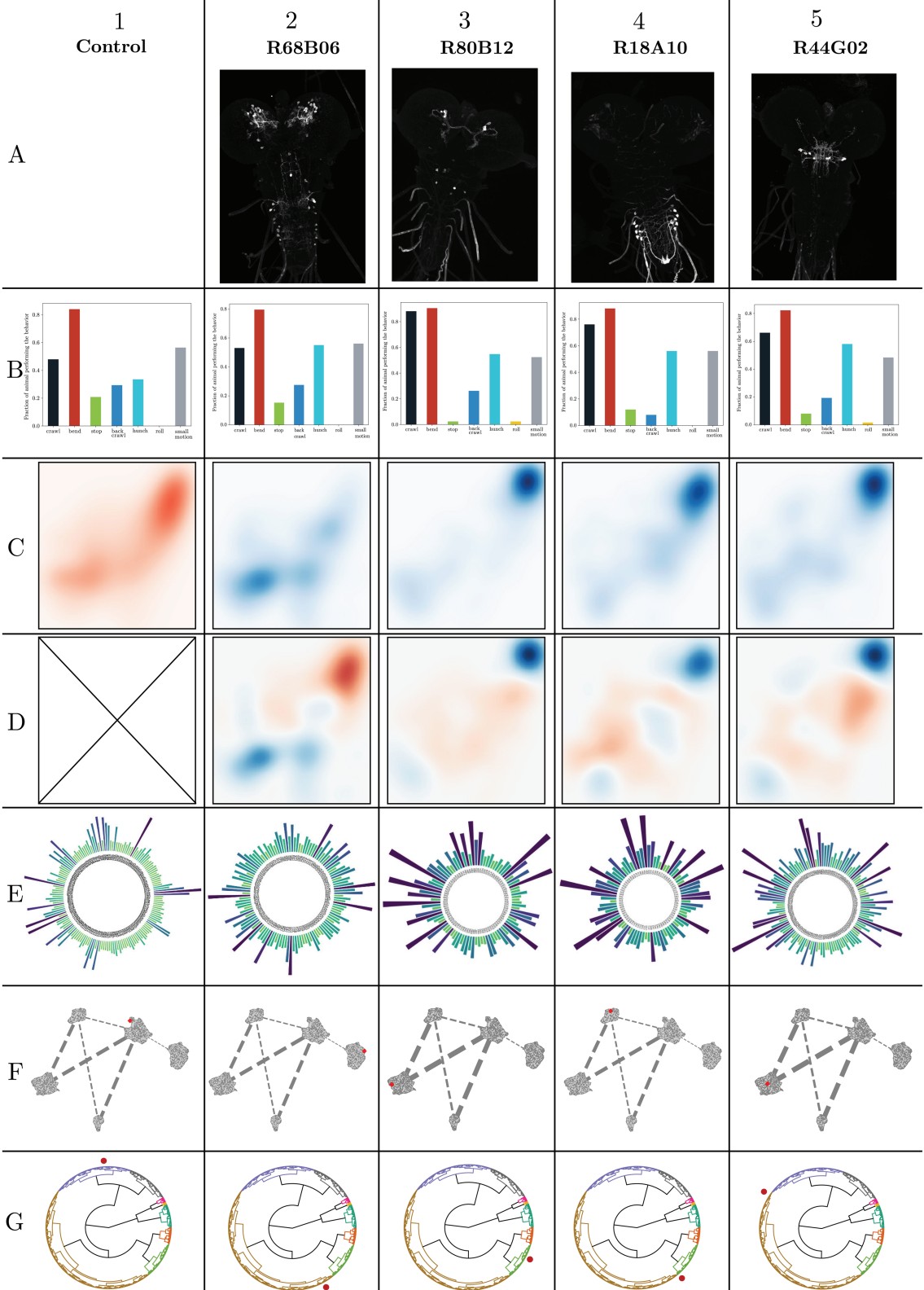

**Fig 7. Samples of genetic lines of interest, "Hits", with their characterization.** These lines lead to subtle modifications in behavior and were not detected by previous approaches. We present four new hits: two hits associated with complex alterations of the learned latent space and two lines associated with significant sequence deviations from the generative model and the reference. The columns

correspond to 1. control line *attP*2, 2. *R68B*06, 3. *R57F*07, 4. *R18A*10, 5. *R38H*09. Row A: Light microscopy images of larval brains expressing the selected GAL4 line. Note that there is no picture for *attP*2, as it is the reference and thus labels no neurons. Row B: Proportions of each stereotypical action evoked during the 2 seconds after stimulis. Row C: Latent distribution of behaviors of the lines, during the 2 seconds after stimulus, with the distribution of the reference line shown in red and the distribution of the hit lines shown in blue. Row D: Witness function between latent distributions of the hit and reference lines, highlighting the main sources of behavioral differences. Note the complex patterns in the latent space, showing these hits do not stem from simple variations in one action. Row E: Z-scores of sequences of three actions between the generative model and experimental sequences. Row F: Position of the reference and hit lines in the 2D representation of the geometric relationships between lines encoded by the distance matrix (shown in Fig 6D,E). Row G: Position of the line in the hierarchical clustering tree (shown here in circular form).

MMD test in the learned latent space and two others identified by comparison with our generative model. Both techniques have considerably broadened the spectrum of lines of interest (i.e. hits) by their capacity to pinpoint behavioral changes that are not overtly manifested by significant changes in individual actions as identified in [33]. Accordingly, each method uncovers distinct sets of characteristics. In particular, intricate patterns in the witness function landscape distinguish the lines newly identified by the MMD method (Fig 7D), indicating alterations across multiple behavioral domains, and new hit lines identified with the generative model correspond to particular z-score profiles (Fig 7E). However, not all modifications in the latent space align with the behaviors defined by the discrete action dictionary.

The new lines identified by the generative probabilistic model are characterized by long-term effects on action sequences, as illustrated in Fig 7E (these lines are listed in S6 Table). Our findings reveal that these lines display variations in the global proportions of specific sequences of three consecutive actions despite having average probabilities of individual actions comparable to the reference line. For sequences beyond three actions, the statistical significance of the findings could not be guaranteed across all lines with the amount of data available, so we restricted our analysis to third order sequences. The newly detected lines were discovered across a broad range of the screen in clusters defined by either the suffix tree representation or the MMD-based distance matrix, as shown in Fig 7F and Fig 7G.

Our methods successfully identified nearly all the hits previously reported by Masson et al. (2020) [33] as strong hits (S5 Table). However, some hits (listed in S7 Table) are no longer classified as such according to the more stringent criteria of our two new approaches. There are several factors contributing to their reclassification. In many instances, transitioning to a definition of behaviors within a continuous latent space (and away from discrete categorization of behavior) eliminates strict boundaries, leading to a loss of significance under the current methodology. It is important to note that different conceptualizations of behavior may yield varying criteria for significance. An increase in the sample size of larvae from these lines will be crucial in determining whether they still qualify as hits under these revised definitions.

In this study, we extended the behavioral paradigms to include a subset of lines (referenced in S8 Table) and examine their behavioral responses to varying levels of air puff intensity. As previously reported, larvae exhibit different behaviors in response to lower stimulus intensities, such as fewer hunches, bends, and backups, and an increase in stops and crawls [33] (Fig 2). Fig 8 presents two lines demonstrating distinct phenotypic variations in their modulation of behavioral responses to different air puff intensities. The first line, R68B05 (Fig 8, column 2), shows a phenotypic difference between the intensities—displaying more pronounced hunching in response to high intensity and less at low intensity than the reference. This results in a greater disparity in hunch response probabilities between low and high stimuli compared to the control, suggesting that neurons in this line may play a role in modulating control and maintaining a stable range of behavioral responses regardless of stimulus

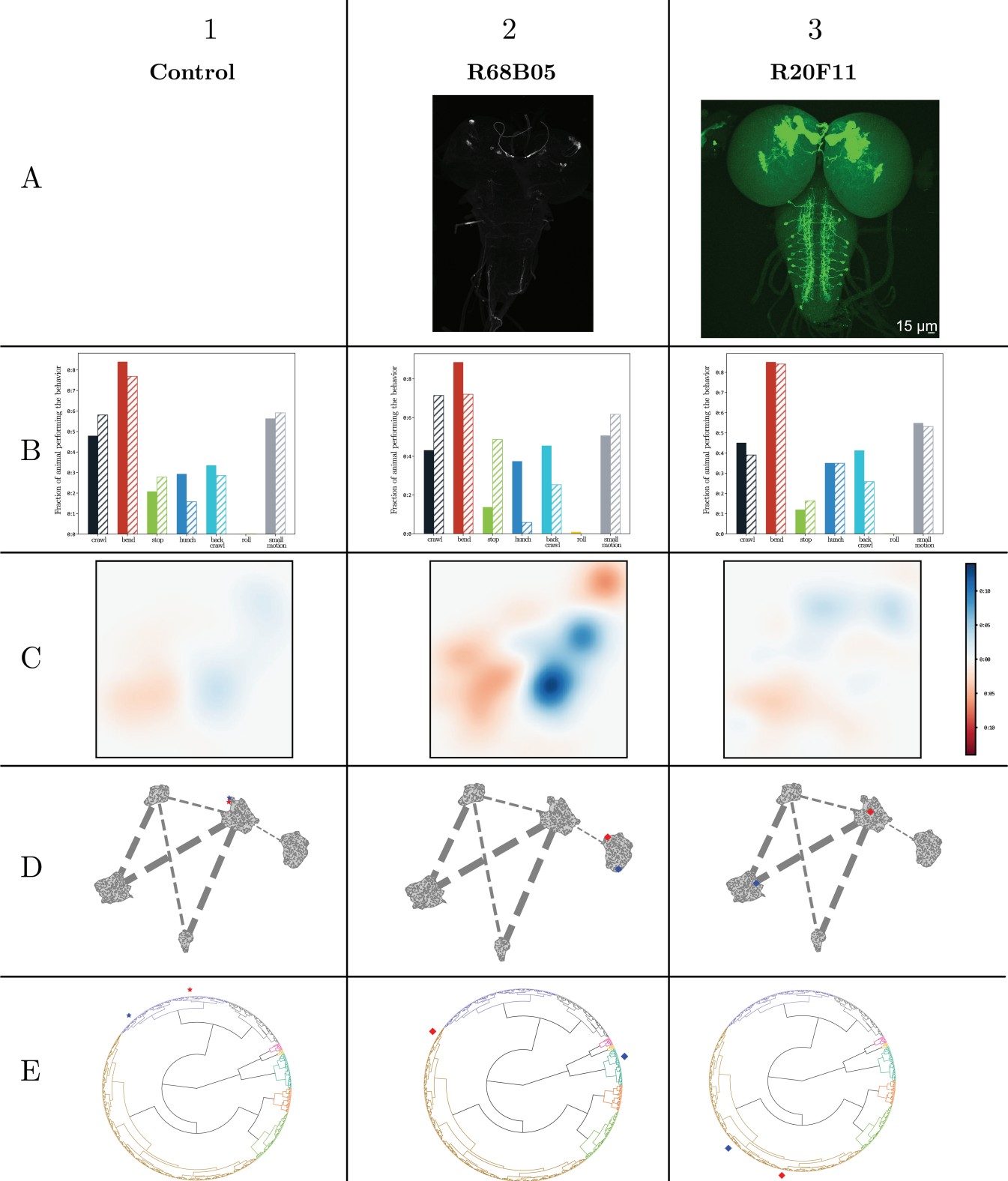

**Fig 8. Two genetic lines of interest, each subjected to two different stimulus intensities: high intensity as previously illustrated, and low intensity, involving a less powerful air puff.** We provide characterizations of each line and protocol. The columns correspond to (1) the control line, (2) *R68B05* and (3)

*R*20*F*11. Row A displays light microscopy images of larval brains expressing the selected GAL4 lines. In Row B, the proportions of each stereotypical action during the 2 seconds following the stimulus, with high intensity in plain color and low intensity in dashed lines. Row C, witness function between latent distributions, highlighting the main sources of behavioral differences between the two protocols for the control and the two lines. Row D, the position of high intensity in red and low intensity in blue in the 2D representation of the geometric relationships between lines, encoded by the distance matrix (as shown in Fig 6B, C). Row E, the positions of high intensity (red) and low intensity (blue) displayed in the hierarchical clustering tree (presented here in circular form)

intensity. The second line, R20F11 (Fig 8, column 2), exhibits a consistent phenotype across both intensity levels, indicating an absence of behavioral modulation based on stimulus intensity (Fig 8A). Neurons in these lines might thus be implicated in encoding stimulus intensity and/or regulating the behavioral response in a stimulus-intensity-dependent manner. Our new methods further support this phenotypic distinction; the witness function uncovers a significant difference in response to high versus low intensity for one protocol compared to the reference, whereas the other displays minimal variation. We can subsequently locate the positions of these two protocols within the latent space and the suffix tree of action sequences.

## Discussion

The swift progress in large-scale behavioral studies, complemented by neural manipulation and recordings, paves the way for establishing causal connections between behavior and its neural underpinnings. Although various statistical methods can identify immediate and pronounced deviations, detecting subtler variations has remained challenging. These minor deviations may stem from nuanced behavioral changes that are difficult to detect or from modulations happening across timescales that are challenging to capture.

We here developed two novel approaches for detecting such subtle modulations. In the first, we represented the dynamics of a population of larvae over a set time scale as a distribution within a continuous, learned latent space based on a compact neural network, trained using a self-supervised objective. Thanks to the low dimensionality of our latent space, we could employ a robust kernel-based statistical test that outperformed previous dictionary-based projections on the detection of subtle behavioral responses. This approach thus allowed us to push automated behavior detection to the frontier of what is ambiguous to human experts. Hence, the complex patterns exhibited by the witness function (examples shown in Fig 7C) for these lines of interest often include the boundaries between previously described discrete behaviors (Fig 3).

The self-supervised approach is adaptable to various architectures, enabling the creation of a meaningful, continuous latent space. It can also be integrated with different interpretations of what constitutes a behavior or an action. Thus, approaches looking for underlying behavioral structures in the spatiotemporal dynamics of postural movement data, [20,23,25], in the latent structure of animal motion prediction [22], or in continuous latent spaces compressing raw video of behavior [26], could be directly patched into our procedure. In this context, the primary practical limitation is that the latent space should be sufficiently low dimensional to allow subsequent statistical testing. As the dimensionality of the latent space increases, so does the risk of anomalies [66–70], a phenomenon commonly known as the curse of dimensionality.

The approach could be extended to larger timescales simply by encoding longer epochs into multiple points, then defining a distribution on the latent space and comparing conditions using the same MMD-test. However, since the ordering of these points would not be represented, long-time window encoding will lose resolution in the temporal sequences of action and, thus, part of the relevant information. Although the MMD exhibits some resilience against the escalation in data dimensions [57,71], the finite number of genetic lines

and corresponding larvae implies that a low-dimensional latent space is necessary to ensure statistical significance of the analysis.

The neural circuitry underlying individual actions and sequences of actions remains only partially understood. There are ongoing debates [72–79] regarding the mechanisms of action initiation, temporal stability, and the transition into new actions. These discussions primarily revolve around whether these processes are localized in specialized centers with centralized competition or are distributed throughout the nervous system. Similarly, there is active debate about the control mechanisms governing sequences of actions, with models such as chains of disinhibitory loops [9], parallel queuing [80], ramp-to-threshold [81,82], and synaptic chains [83,84] under consideration. Our second approach focuses on uncovering complex correlations within the structure of action sequences at the population level. Although behavior alone may not conclusively pinpoint the neural mechanisms responsible for generating sequences, complexities observable at the population scale, such as higher-order correlations or other non-Markovian characteristics, may offer clues about the neurons that orchestrate these dynamics in sequence generation.

In our second approach, we used the structured framework of a tractable probabilistic generative models to explore complexities in action sequences. This model is a foundation for contrasting a group of larvae against a corresponding constrained model, eliminating the need for an external reference line for comparison. Our method is adept at identifying complex temporal variations in sequences at the population level, due to our analysis of higher-order correlations within these sequences and comparing them against the constraints of the generative model. In our model, the future state of an individual is determined solely by its current state, i.e., it is a Markov model. The model also incorporates variability through a potentially time-varying effective action rate, i.e., the model is time-inhomogeneous. The model does not explicitly account for the frequencies of sequences of three or more consecutive actions. Thus, it enabled the identification deviations from Markovianity of such three-action sequences and let us construct a z-score (or, equivalently, significance) profile for individual genetic lines, describing to which extent the time-evolving probabilities of actions alone describe experimental behavior sequences. This makes it possible to identify "hit" lines whose action sequences are significantly non-Markovian, and it provides an efficient metric for comparing the behavioral dynamics of different genetic lines.

The intricacies of behavior and its connection to neural computations, whether in specialized circuits or distributed across the nervous system, cannot be fully understood through a single method or confined to a particular time scale. However, we can identify meaningful behavioral characteristics by integrating multiple methods that utilize both local and global data at the level of individual animals and across populations and by spanning various time scales. These characteristics can then be compiled into behavioral identity documents for individual neurons and neuronal clusters (Fig 7). By merging these detailed profiles with connectome data [3] and neural recordings, we can advance the discovery of circuits responsible for decision-making and the subtle nuances in their output.

## Supporting information

**S1 Text.** Supporting information on the models described in the **Methods** section.
(PDF)

**S1 Video.** Video version of Fig 2C.
(GIF)

**S1 Table.** List of lines studied.
(CSV)

**S2 Video.** Video version of Fig B in S1 Text.
(MP4)

**S2 Table.** Estimated square of the latent space Maximum Mean Discrepancy (MMD) with the corresponding bootstraped p-values for all lines, complemented by a calculation of the distance between generative and experimental sequences.
(CSV)

**S3 Table.** Z scores comparing action sequences from the generative model to experimental recordings obtained respectively during the stimulus and at all times.
(CSV)

**S4 Table.** Z scores comparing action sequences from the generative model to experimental recordings obtained respectively during the stimulus and at all times.
(CSV)

**S5 Table.** Estimated square of the latent space Maximum Mean Discrepancy (MMD) and the distance between generative and experimental sequences for lines previously identified as hits, as cited in [33], and confirmed as such with this analysis.
(CSV)

**S6 Table.** List of new hits detected by the two new approaches.
(CSV)

**S7 Table.** Genetic lines that were detected as hits in [33] and that no longer are hits with this analysis.
(CSV)

**S8 Table.** Estimated square of the latent space Maximum Mean Discrepancy (MMD) with the corresponding bootstraped p-values, complemented by a calculation of the distance between generative and experimental sequences for lines with low and high intensity of the stimulus.
(CSV)

## Acknowledgments

We acknowledge the help of the HPC Core Facility of the Institut Pasteur for this work.

## Author contributions

**Conceptualization:** Alexandre blanc, François Laurent, Alex Barbier–Chebbah, Marta Zlatic, Rayan Chikhi, Christian L. Vestergaard, Tihana Jovanic, Jean-Baptiste Masson, Chloé Barré.

**Data curation:** Alexandre blanc, François Laurent, Benjamin T. Cocanougher, Benjamin M.W. Jones, Peter Hague, Tihana Jovanic, Chloé Barré.

**Formal analysis:** Alexandre blanc, François Laurent, Christian L. Vestergaard, Tihana Jovanic, Jean-Baptiste Masson, Chloé Barré.

**Funding acquisition:** Christian L. Vestergaard, Tihana Jovanic, Jean-Baptiste Masson.

**Investigation:** Alexandre blanc, François Laurent, Alex Barbier–Chebbah, Hugues Van Assel, Christian L. Vestergaard, Tihana Jovanic, Jean-Baptiste Masson, Chloé Barré.

**Methodology:** Alexandre blanc, François Laurent, Alex Barbier–Chebbah, Marta Zlatic, Rayan Chikhi, Christian L. Vestergaard, Tihana Jovanic, Jean-Baptiste Masson, Chloé Barré.

**Project administration:** Tihana Jovanic, Jean-Baptiste Masson.

**Resources:** François Laurent, Benjamin T. Cocanougher, Benjamin M.W. Jones, Peter Hague, Marta Zlatic, Tihana Jovanic, Chloé Barré.

**Software:** Alexandre blanc, François Laurent, Hugues Van Assel, Chloé Barré.

**Supervision:** François Laurent, Marta Zlatic, Christian L. Vestergaard, Tihana Jovanic, Jean-Baptiste Masson, Chloé Barré.

**Visualization:** Alexandre blanc, François Laurent, Chloé Barré.

**Writing – original draft:** Alexandre blanc, Alex Barbier–Chebbah, Marta Zlatic, Christian L. Vestergaard, Tihana Jovanic, Jean-Baptiste Masson, Chloé Barré.

**Writing – review & editing:** Alex Barbier–Chebbah, Marta Zlatic, Christian L. Vestergaard, Tihana Jovanic, Jean-Baptiste Masson, Chloé Barré.

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
