## [Decision Letter · Decision Letter 0]

1 Oct 2024

Dear Dr Masson,

Thank you very much for submitting your manuscript "Statistical signature of subtle behavioural changes in large-scale behavioural assays." for consideration at PLOS Computational Biology. As with all papers reviewed by the journal, your manuscript was reviewed by members of the editorial board and by an independent reviewer. The reviewers appreciated the attention to an important topic. Based on the reviews, we are likely to accept this manuscript for publication, providing that you modify the manuscript according to the review recommendations. Please make sure that the code is in good shape. We will likely move this paper to the methods section

Sincerely,

Daniele Marinazzo

Section Editor

PLOS Computational Biology

Daniele Marinazzo

Section Editor

PLOS Computational Biology

Reviewer's Responses to Questions

**Comments to the Authors: 
Please note here if the review is uploaded as an attachment.**

Reviewer #1: Blanc et al extend prior work by Masson and colleagues in automating the quantitative analysis of Drosophila larval behavioral dynamics from video microscopy. The intelligent use of a simple physical model of larval shape -- not much more than the physics of an elastic larva-shaped object -- is a principled way to regularize the extraction of larval posture from video. Temporal continuity is incorporated into the analysis in a straightforward way using an autoencoder that is fed multiple successive time points. The system seems like it robustly assigns behavioral categories to every time point, and effectively assigns quantitatively-rich behavioral phenotypes spanning state probabilities and transition rates.

The field now needs toolboxes like this one. Large Drosophila libraries with well-defined genetic perturbations demand automated methods for quantitative phenotyping. As the field moves more deeply into higher-order behavioral decision-making, these phenotypes are likely to be more nuanced. Quantitatively significant but visibly subtle differences between animals may harbor important insights into the workings of brain circuits.

I have no major suggestions. The paper is highly technical, but generally reads well. There are *many* typos and misspellings. The entire paper needs careful proofreading. I didn't try to proofread the equations. I only skimmed the equations for general plausibility. The authors would do well to go through the entire manuscript with a fine-toothed comb.

**Have the authors made all data and (if applicable) computational code underlying the findings in their manuscript fully available?**

Reviewer #1: Yes

PLOS authors have the option to publish the peer review history of their article (what does this mean?). If published, this will include your full peer review and any attached files.

Reviewer #1: No

Figure Files:

Data Requirements:

Reproducibility:

References:

---

## [Editor Report · Decision Letter 1]

24 Mar 2025

Dear Dr Masson,

We are pleased to inform you that your manuscript 'Statistical signature of subtle behavioral changes in large-scale assays' has been provisionally accepted for publication in PLOS Computational Biology.

Best regards,

Daniele Marinazzo

Section Editor

PLOS Computational Biology

Daniele Marinazzo

Section Editor

PLOS Computational Biology

---

## [Editor Report · Acceptance letter]

PCOMPBIOL-D-24-01025R1

Statistical signature of subtle behavioral changes in large-scale assays

Dear Dr Masson,

I am pleased to inform you that your manuscript has been formally accepted for publication in PLOS Computational Biology. Your manuscript is now with our production department and you will be notified of the publication date in due course.

With kind regards,

Anita Estes
